# Recyclization of Maleimides by Binucleophiles as a General Approach for Building Hydrogenated Heterocyclic Systems

**DOI:** 10.3390/molecules27165268

**Published:** 2022-08-18

**Authors:** Dmitriy Yu. Vandyshev, Khidmet S. Shikhaliev

**Affiliations:** 1Department of Organic Chemistry, Faculty of Chemistry, Voronezh State University, Universitetskaya Sq. 1, 394018 Voronezh, Russia; 2TekhnoKhim, 50 Let Sovetskoi Vlasti Str. 8, 394050 Voronezh, Russia

**Keywords:** *N*-arylmaleimides, recyclization, hydrogenated heterocyclic compounds

## Abstract

The building of heterocyclic systems containing hydrogenated fragments is an important step towards the creation of biologically-active compounds with a wide spectrum of pharmacological activity. Among the numerous methods for creating such systems, a special place is occupied by processes using *N*-substituted maleimides as the initial substrate. This molecule easily reacts in Diels-Alder/retro-Diels-Alder reactions, Michael additions with various nucleophiles, and co-polymerization processes, as have been described in numerous detailed reviews. However, information on the use of maleimides in cascade heterocyclization reactions is currently limited. This study is devoted to a review and analysis of existing literature data on the processes of recyclization of *N*-substituted maleimides with various *C*,*N*-/*N*,*N*-/*S*,*N*-di- and polynucleophilic agents, such as amidines, guanidines, diamines, aliphatic ketazines, aminouracils, amino- and mercaptoazoles, aminothiourea, and thiocarbomoyl pyrazolines, among others. The significant structural diversity of the recyclization products described in this study illustrates the powerful potential of maleimides as a building block in the organic synthesis of biologically-active compounds with hydrogenated heterocyclic fragments.

## 1. Introduction

Fragment-based drug discovery (FBDD) is a well-established method for creating new hits and leads [1,2,3,4,5,6]. This approach has been repeatedly confirmed in practice and it is an additional strategy to supplement other search methods, such as high-throughput screening [7].

A detailed evaluation of many existing “fragmentary” libraries indicates the predominance of (hetero)aromatic “planar” compounds and the very low diversity of chiral compounds rich in C*sp*^3^ atoms [8,9].

However, studies by Ritchieetal [10] and Loveringetal [11] demonstrated that an increase in the proportion of C*sp*^3^ atoms in a molecule or the limitation of the number of aromatic rings significantly increase the activity of compounds and their easy passage through barriers. In addition, it has been proven that mainly systems rich in C*sp*^3^ atoms are used in real clinical practice after human trials [12]. All these facts point to the necessity to build unsaturated condensed or linear coupled ensembles, both for the development of screening collections and in the subsequent development of hit to lead.

The number of studies aimed at synthesizing heterocyclic compound collections enriched in C*sp*^3^ atoms is very limited, and therefore access to new types of scaffolds is limited. There are only a few studies [13] devoted to the development of new approaches and methods for constructing fragments with several synthetically available three-dimensional growth vectors, which provide fast and efficient development of hit to lead after initial screening. Considering all the above points, an important task is the development of efficient synthetic routes for partially saturated bicyclic heteroaromatic (PSBH) frameworks with an increased content of C*sp*^3^ atoms compared to existing libraries.

The requirements for the efficiency of synthetic methods are constantly increasing due to the need to simultaneously increase the molecular complexity and minimize the number of steps in synthetic procedures. Cascade (tandem, domino) processes are a very promising methodology for organic synthesis, allowing the structure of the target molecule to complicate by combining a series of successive transformations in one synthetic operation. Among the developed methodologies for domino transformations, the most effective is the sequence of reactions, at the key stage of which the formation of a heterocyclic system occurs as a result of the recycling of the intermediate. With this approach, the chemo-, regio-, and stereoselectivity of processes usually increases significantly due to the greater determinism of the location of the reaction centres of the reagent and substrate.

The problem of searching for easily accessible, polyfunctional substrates that allow for directed cascade synthesis of various heterocyclic structures is one of the key ones. In this context, *N*-arylmaleimides deserve special attention [14,15]. Their interaction with various reagents, including the domino route, can lead to the formation of a large number of hydrogenated heterocyclic systems. However, only (retro-) Diels-Alder reactions [14,16,17], Michael additions [16,17] with various nucleophiles, and co-polymerization processes [16,18] have been studied in detail to date. This study is devoted to a review and analysis of existing literature data on the processes of recyclization of *N*-substituted maleimides with various linear and cyclic di- and polynucleophilic agents.

The analysis of the available literature data allowed us to draw a conclusion regarding the sequence of these reactions. The initial nucleophilic addition of a di- or polynucleophile according to the Michael reaction to the activated multiple bond of the imide and subsequent recyclization of the intermediate succinimide intermediate proceeds due to intramolecular nucleophilic substitution with the participation of another nucleophilic centre and one of the carbonyl groups. It should be noted that, depending on the structure of the dinucleophilic component and the selected conditions, the formation of various alternative products is possible, which is reflected in Figure 1.

## 2. Reactions with *N*,*N*-dinucleophiles

Partially hydrogenated azoloazines represent a class of heterocyclic compounds with high biological and pharmacological activity [19,20,21]. In addition, these systems are often used as a simple model for studying such fundamental issues of medical and organic chemistry as conformational flexibility, tautomerism, electronic effects, etc. [10,11]. One of the most effective approaches to the building of partially hydrogenated azoloazines is the reactions of 1,3-*N*,*N*- and 1,4-*N*,*N*-dinucleophiles with maleic anhydride and its imides.

A typical example of recyclization reactions of maleimides **1** with 1,3-*N*,*N*-binucleophiles, is their interaction with carboxymidamides **2** (Figure 2). Thus, Kh. S. Shikhaliev [22] and Yu. A. Kovygin [23] et al. found that the optimal conditions for these processes are boiling the mixture of reagents in acetone or chloroform. It was assumed that the mechanism of the process consists of two successive stages. During the first stage, the nucleophilic addition of the carboximidamide amino group at the double bond of the *N*-arylmaleimide molecule occurs. The resulting adduct undergoes subsequent tandem recycling of the succinimide moiety to form substituted 2-[4-oxo-4,5-dihydro-1*H*-imidazol-5-yl]-*N*-arylacetamides **4**, the structure of which was proved using XRD analysis. The authors also noted that when using methanol, isopropanol, water, or dimethylformamide, mixtures which are difficult to separate were formed, which was most likely due to the solvolysis of maleimide that had been catalyzed by the highly basic carboxamide. In addition to the main target compound **4** (the yield fluctuated within 50%), the transamidation product **5** (yield up to 10%) was also isolated. The attempts for cyclization of this product were unsuccessful.

We know about examples of substitution of aliphatic 1,3-binucleophiles in these reactions by their heterocyclic analogues containing a guanidine fragment [24]. Thus, for 2-aminobenzimidazole **7** and 2-aminotriazole **8** when they interact with **1** in dioxane medium, the corresponding 2-oxo-1,2,3,4-tetrahydrobenzo[4,5]imidazo[1,2-*a*]pyrimidine-4-carboxyanilides **9** and 7-oxo-5,6,7,8-tetrahydro[1,2,4]triazolo[4,3-*a*]pyrimidine-5-carboxyanilides **13** were isolated. Taking into account the non-equivalence of nucleophilic centres in the initial heterocyclic matrices, the formation of several regioisomeric products during these processes (**9**–**12**) shown in Figure 3 for 2-aminobenzimidazole is possible. The choice of the structure of the obtained compounds was carried out using the detailed analysis of the literature data [25] and obtained spectral [24,26] data.

Later, R.V. Rudenko et al. [26] showed that varying the solvent significantly changes the direction of this reaction. If in the case of 2-aminobenzimidazole the replacement of dioxane by dimethylformamide only reduced the reaction time, a completely different pattern would be observed for 2-aminotriazole **8**. Thus, when the reaction was performed in the acetic acid or DMF medium, the regioisomer **14** was formed instead of the expected compound **13** (Figure 4).

However, in the case of the 5-amino-4-R-pyrazoles [27] **15** variation of aromatic substituents in **1**, it allowed both tetrahydropyrazolo[1,5-*a*]pyrimidines **16** and dihydroimidazo[1,2-*b*]pyrazoles **17** to be isolated (Figure 5, Table 1). The authors also noted that when the reaction was carried out in isopropyl alcohol, a linearly-bound intermediate **18** was formed. When this intermediate was boiled in acetic acid, the mixture of **16** and **17** was formed, and intermediate **16** was obtained as the result of boiling in dimethylformamide.

Catalysts play a special role in recyclization reactions involving *N*-arylmaleimides. T. Matviyuk [28] found that the interaction of 2-aminopyridines **19** with maleimides **1** in a dioxane medium with the involvement of lithium perchlorate led to the formation of 2-oxo-2,3-dihydroimidazo[1,2-*a*]pyridin-3-yl derivatives of succinmide **24** (Figure 6). The authors suggested that during the first stage, 2-aminopyridine reacts with maleimide at the endocyclic nitrogen atom, forming a substituted succinimide, the subsequent recycling of which leads to intermediate compound **22**. Alternatively, this reaction may proceed with an initial attack of the exocyclic nitrogen atom followed by the formation of adduct **23**. However, the performed NOESY analysis demonstrates a strong correlation between the α-H of the pyridine ring and the proton of the methylene group in the side chain. Further, intermediate adduct **22**, being a strong CH-acid, reacts with the second maleimide molecule, which leads to the final product **24**.

A series of studies was devoted to the investigation of the interaction of maleimides with 1,4-binucleophiles. For 1,2-diaminoethane **25**, *N*,*N*-dibenzylethane-1,2-diamine **26,** and 12,-diaminocyclohexane **27**, which are symmetrical dinucleophiles, A. V. Zorina [29] isolated the following compounds by boiling the mixture of reagents in methanol: 2-(3-oxopiperazin-2-yl)-acetanilides **28**, 2-acet-4-methylanilido-1,4-dibenzyl-3-oxopiperazines **29,** and 2-acetanylido-3-oxodecahydroquinoxalines **30** (Figure 7).

M. M. Abelman [30] noted that when using unsymmetrical ethylenediamines, the reaction in ethanol at room temperature, depending on the radicals, can lead to various recyclization products (Figure 8, Table 2).

In the case of reaction **1** with o-phenylenediamine [31] (Figure 9), regardless of the solvent used, the formation of only tetrahydroquinaxolinyl acetanilides **37** was noted. At the same time, the maximum yield of end products (67%) was achieved in aqueous ethanol.

## 3. Reactions with *S*- and *O*-containing Dinucleophiles

The thiazole molecule is a good pharmacophore nucleus due to its various pharmaceutical applications. Its derivatives possess a wide spectrum of biological activity, such as antioxidant, analgesic, antibacterial, anticancer, anti-allergic, antihypertensive, anti-inflammatory, antimalarial, antifungal, and antipsychotic effects [32,33,34,35,36,37].

For the first time, the possibility of using *N*-arylmaleimides to build a thiazole ring was described by M. Augustin [38] and D. Marrian [39]. Thiourea and *N*-phenylthiourea, which are examples of 1,3-*N*,*S*-dinucleophiles, were considered as initial substrates. The reaction was carried out in a dioxane medium (Figure 10).

The obtained results allowed the extension of this reaction to other systems containing a thioamide fragment (Figure 11 and Figure 12). Thus, interactions of primary and secondary thioamides with *N*-arylmaleimides were considered by T. Takido [40]. It has been established that, regardless of the structure of the thioamide component, the optimal conditions for the reaction are boiling the mixture of reagents in a dioxane medium. It was assumed that this reaction proceeds due to the nucleophilic attack by the sulphur atom of the thioamide **42** at the double bond of *N*-arylmaleimide **1** with the formation of succinimide **43**. The intramolecular attack of the imine nitrogen on the nearest carbonyl group and subsequent recyclization of the imide ring leads to mesoionic intermediate **44**. Further, due to the intramolecular rearrangement of the proton or alkyl (aryl) group from the nitrogen atom to the oxygen atom, the formation of final thiazoles **45** occurs.

Later, D. Havrylyuk [41] et al. extended this process to heterocyclic systems containing a thioamide component, in particular, to 3-phenyl-5-aryl-1-thiocarbamoyl-2-pyrazolines (Figure 12).

Interesting and contradictory results have been obtained in the case of heterocyclic matrices, in which the thioamide fragment was part of the cyclic system (Figure 13). Lesyk R. et al. [42], as a result of boiling an equimolar mixture of reagents for 2 h in acetic acid, were able to isolate *N*-(R-phenyl)-(6-oxo-5,6-dihydro[1,3]thiazole[3,2-*b*][1,2,4]triazol-5-yl)acetamides **51** based on the example of 1,2,4-triazole-3(5)-thiol **50**. However, S. Holota [43], when investigating the same interaction, showed that boiling an equimolar mixture in the range of 30 min–24 h and the use of such solvents as acetic acid, acetone, acetonitrile, benzene, and toluene in the presence or absence of sodium acetate only led to the formation of linearly-bound product **52**. At the same time, attempts at the cyclization of **52** did not lead to success. Such a discrepancy in the obtained experimental results is due to the fact that in [42], the conclusion about the structure of the obtained compound was made only based on the interpretation of the spectral data. Thus, the signals in the ^1^H NMR spectra at δ = 13.8 and -14.3 ppm were treated as NH protons in amide group **51**. However, a similar singlet in a similar magnetic field can belong to the signal of NH of the proton of triazole ring **52.** This controversial issue was resolved in [43] by analyzing XRD data.

Another direction in the chemistry of *N*-arylmaleimides, which is well represented in the literature, is the building of thiomorpholine rings when they interact with 1,4-*N*,*S*-binucleophiles [38,44] (Figure 14). This framework is a common pharmacophoric element and it exhibits selective enzyme inhibition for many receptors and other types of molecular targets [45,46,47,48,49,50,51].

The reactions of maleimides with the involvement of 1,4-*N*,*O*-binucleophiles also proceed in a similar way. During these transformations, unsaturated oxazine (morpholine) cycles are formed, which are part of a number of compounds with a wide spectrum of biological activity [52,53,54,55,56,57,58,59,60,61,62,63,64,65,66]. A. V. Zorina et al. [67,68] studied the interaction of **1** with aminophenol **55** and aminoethanols **56** in detail. It was noted that both linearly-bound adduct **57** and cyclic regioisomers **58** and **59** can be isolated when **55** was introduced into the reaction, which varies the solvents and catalysts (Figure 15).

The scientific team led by I. Ito [69] obtained succinimide **62** in a similar interaction using unsubstituted maleimide and 2,4-diamino-5-hydroxy-6-methylpyrimidine (Figure 16) in ethanol medium (the yield was 90%). It has been shown that boiling **62** in water resulted in hydrolysis with the formation of *β*-carbamoyl-(2,4-diamino-6methylpyridin-5-yl)hydroxyethylcarboxylic acid **63** and **64**. If **63** was treated with sodium acetate in a mixture of acetic acid and ethanol, 2-acetamido-8-acetyl-4,6-dimethyl-6*H*-pyrimido[5,4-*b*][1,4]oxazin-7-one was formed, which can also be obtained via an alternative pathway (Figure 17). The substance, 2-amino-6-carbamoylmethyl-4-methyl-6*H*-pyrimido[5,4-*b*][1,4]oxazin-7(8*H*)-one **66,** was formed when succinimide **62** was heated in ethanol with the addition of catalytic amounts of triethylamine.

Interestingly, when using *N*-arylmaleimides, the process proceeds in a similar way, however, the authors were not able to isolate intermediate **71** (analogue **62**) (Figure 18).

## 4. Reactions with *C*,*N*-dinucleophiles

Recyclization reactions of maleimides during their interaction with *C*,*N*-binucleophiles occupy a special place in the literature. Depending on the structure of the binucleophilic component, such processes can form partially hydrogenated pyrrole, pyridazine, or pyridine fragments.

The 2-pyrrolidine core is one of the most abundant structural fragments in natural compounds and is also an important intermediate in the development of new drugs [70,71,72,73,74,75,76]. Among the entire array of data, a special place in the synthesis of this framework is occupied by the proposed Yu. A. Kovygin et al. [23] interaction of **1** with *β*-aminocrotonic acid methyl ester **74** as a representative of 1,3-*C*,*N*-binucleophiles. The process was carried out under various conditions: boiling in organic solvents (diethyl ether, alcohols, dioxane, dimethylformamide, acetic acid), including using acidic or basic catalysis. The authors found that, regardless of the medium used, in all cases the same major product of 5-oxo-4,5-dihydro-1*H*-pyrrol-3-carboxylate **76** is formed (Figure 19). However, monitoring of the reaction showed that the maximum yield of 75–80% can be achieved by boiling the starting reagents in methanol with the addition of catalytic amounts of toluenesulfonic acid. In this case, the chemical route of the reaction assumes, as in the previous cases, the initial nucleophilic Michael addition of the CH-proton of the enamine molecule at double bond **1**. Further, via the imino-enamine tautomerism stage, the amino group of **74** is attacked by carbonyl moiety **1**, which is accompanied by the opening of the pyrrolidinone ring and the formation of a new pyrroline ring.

Another promising direction in the chemistry of maleimides, which involves the use of 1,3-*C*,*N*-binucleophiles, is the building of subunits of 4,5-dihydropyridazin-3(2*H*)-one. It is known that this fragment is present in compounds with a significant spectrum of biological activity: phosphodiesterase 3/4 (PDE3/PDE4) inhibitors, [77] cyclooxygenase-2 (COX-2) inhibitors, [78] subtype-4 receptor agonists (MC4R), [79] platelet aggregation inhibitors, [80] adenosine-3′,5′-cyclic phosphate phosphodiesterase III (CAMPPDEIII) inhibitors, [81] p38 MAP kinase inhibitors, and [82] *β*-adrenergic antagonists, [83] and in compounds with antihypertensive, [84] positive inotropic, [85] cardiotonic, [86] antithrombotic, anti-inflammatory, and anti-ulcer effects [87]. A. Stepakov [88] showed that a simple and convenient way to build 4,5-dihydropyridazin-3(2*H*)-ones is the interaction of aliphatic and cyclic ketazines with arylmaleimides (Figure 20).

Although the mechanism of this reaction is not understood, the authors propose a probable route involving the 1,4-addition of the tautomeric form of the azine **A** to maleimide **1** with the formation of product **B** of the Michael addition. Further, adduct **B** can be converted into 4,5-dihydropyridazine-3(2*H*)-one **D** via *N*-substituted dihydropyridazinone **C**. Formation of dihydropyridazinone **C** occurs by intramolecular nucleophilic substitution with a carbonyl group (Figure 21).

Fused heterocyclic systems containing a partially hydrogenated pyrimidine ring have already proved to be efficient in medicinal chemistry and are promising objects for the creation of new drugs [89,90,91,92,93,94,95].

The authors of several studies [96,97,98] showed that Michael adducts are formed as a result of the interaction of 6-aminouracils **84** with *N*-arylmaleimides **1** in acetonitrile or isopropyl alcohol medium. However, the scientific team of R. Rudenko [99] succeeded in isolating the corresponding recycling products by changing the process conditions to boiling the reagents in acetic acid. It should be noted that, depending on the substituents in the structure of 6-aminouracils, the authors were able to isolate the corresponding succinimide **86**, and fused pyridopyrimidines **85** and pyrrolidinopyrimidine **88** (Figure 22). It was found that in the case of unsubstituted **84**, the replacement of acetic acid—boiling of which led to the formation of mixture **85** and **86**—with DMF led to the formation of single product **85**. In this case, complete conversion was achieved after only 3 h of boiling of an equimolar mixture of reagents. Taking into account the fact that resulting systems **85′** and **88** have the same set of signals in ^1^H and ^13^C spectra, their structures were proved using NOE and XRD experiments.

P. Romanov [100] showed that 2,4,6-triaminopyrimidines behave in a similar way (Figure 23).

Among the large array of literature data on the building of bicyclic pyrrolidinones [101,102,103,104,105], the reactions of *N*-arylmaleimides with heterocyclic ketene amines (HKAs) deserve special attention [106] (Figure 24). It was found that the optimal conditions for carrying out this process is a 20-min stirring of the mixture of reagents in an ethanol medium, while the yield of final bicyclic pyrrolidinones reached 85%. The mechanism of this reaction is similar to the interaction of **1** with *β*-aminocrotonic acid methyl ester **74** shown in Figure 18.

Another example of the formation of pyrimidine and pyrrole rings is the interaction of *N*-arylitaconimides with 1,3-substituted 5-aminopyrazoles [27] **95** (Figure 25). The process conditions were similar to those for 6-aminouracils. NOESY and XRD experiments were also used to establish the structure of compounds **96** and **97**.

## 5. Reactions with Polynucleophilic Reagents and Involvement in Multicomponent Processes

In conclusion, we should consider the interaction with polynucleophiles as the most promising and least studied direction currently in the recyclization reactions of maleimides. Now, such processes have been well studied using the example of *N*,*N*-, *C*,*N*- and *N*,*S*-containing polynucleophilic agents. In addition to the alternative opening of the maleimide ring, polynucleophiles with several non-equivalent reaction centres can contribute to the formation of various fused or linearly-linked systems, as well as their mixtures.

Despite the fact that aryl biguanidines **98** are polynucleophilic compounds, A.V. Zorina [107] and Yu.A. Kovygin [23] found that, when they interact with *N*-arylmaleimides, cyclization occurs during boiling in methanol with the involvement of the guanidine moiety only (Figure 26). Thus, biguanidines behave like typical 1,3-*N*,*N*-dinucleophiles. It is also worth noting that attempts to change the conditions and the use of acidic or basic catalysis did not affect the change in the reaction route. The structure of the resulting 5-oxo-4,5-dihydroimidazol-4-yl-*N*-arylacetamides **99**, for which the existence of a tautomeric form **99′** is also possible, was proved using the NOESY and XRD spectra. When polynucleophiles containing simultaneously competing 1,3-*N*,*N*- and 1,3-*N*,*S*-dinucleophilic centres were introduced into such reactions, the situation became more complicated. However, using the example of amidinothiourea **100** and thiosemicarbazone **101,** it was established [107,108] that the reactions proceed at 1,3-*N*,*S*-dinucleophilic centres, forming the corresponding thiazoline cycles **102** and **103**.

As mentioned earlier, the thiazoline ring can also be obtained from thioureas and *N*-phenylthioureas, the reaction of which with maleimides proceeded unambiguously. However, in the case of *N*,*N*-substituted thioureas, the reaction can proceed via several alternative routes (Figure 27). Thus, the study of A. S. Pankova [109] showed that the reaction of N-alkylthioureas with maleimides at room temperature in ethanol can lead to the formation of mixture **104** and **105**. In addition, the nature of the solvent, substituents, and the steric factor play an important role in the formation of specific regioisomers [110]. This result, according to the authors, is due to the fact that when using polar solvents, the reaction is subject to kinetic control, while non-polar solvents contribute to the formation of structure **108**, which is more thermodynamically favourable.

L. Salhi et al. [108] found that 2,3-diaminopyrimidine in the reaction with maleimides acts as a 1,3-*N*,*N*-dinucleophile, forming imidazo[1,2-*a*]pyridines with a wide range of biological activity [111,112,113,114]. As in [23], the authors suggested that the formation of **115** occurs first as a result of nucleophilic attack of the cyclic nitrogen atom of 2,3-diaminopyridine **110** of the double bond of maleimide **1**. Further opening of fragment ring **1** occurs due to the intramolecular attack of the imine nitrogen on the carbonyl group, which was previously protonated with acid, with the formation of intermediate product **112**. Next, second molecule **110** acts as a base and removes an acidic proton Hx with the formation of the final product according to Knoevenagel (Figure 28).

Later, it was shown [115] that 1,2-daiminobenzimidazole, which contains competing 1,4-*N*,*N*- and 1,3-*N*,*N*-dinucleophilic fragments, exclusively reacts with maleimides as 1,3-*N*,*N*-dinucleophile (Figure 29). It should be noted that, unlike diaminopyridine, this reaction proceeds via the formation of succinimide by the aza-Michael reaction due to the exo-amino group in the second position.

The interactions of maleimides with aminoazoles also proceed ambiguously. R.V. Rudenko [27] showed that in the reaction of 5-amino-2-R-pyrazoles with **1,** regardless of the solvent used, a mixture of pyrazolopyrimidines **119** and pyrazolopyridines **126** was formed (Figure 30). However, for such polynucleophiles as 1,2-diamino-4-phenylimidazole [116], it was possible to isolate only one of the possible products—imidazodiazinon **122**. In this case, the replacement of the solvent only affected the yield of the final product. The authors suggested that at the first stage, diaminoimidazole is added to the double bond of arylmaleimide **1** with the formation of linearly-bound products due to CH of the imidazole cycle or NH_2_ groups. Further, intramolecular cyclization of the resulting intermediates can lead to various alternative products. The exact structure of the resulting imidazopyridazine **122** was established by the step-by-step reaction with the release of succinimide **121**.

Other promising but poorly studied issues relating to the chemistry of maleimides are multicomponent processes. At the moment, there are only a few examples of such reactions.

The first three-component synthesis involving maleimides was described by T. Takido [40] using the example of the interaction of **1** with thioamides and maleic anhydride **123** (Figure 31), which led to rather unexpected tricyclic bridge systems. The optimal conditions for this process are the 3-h boiling of a mixture of reagents in dioxane. The mechanism proposed by the authors includes the first stage similar to that shown in Figure 11, which consists of the formation of succinimides **124** and **43** due to nucleophilic attack by the sulphur atom of thioamide **42** at the double bond of *N*-arylmaleimide **1** or maleic anhydride **123**. Further recycling of **124** and **43** lead to the formation of mesoionic intermediates **44** and **125**, which easily enter into 1,3-dipolar cycloaddition reactions with **1**- and **123**-forming final products—tricyclic bridge systems **126–128**.

The second example of a multicomponent process was proposed by J. Noth [117] and E. Fenster [118] for the production of *γ*-lactams (Figure 32) as a result of the interaction of maleimides, aldehydes, and amines in the presence of reducing agents. The route of this cascade reaction includes the formation of formyl methylsuccinimide **130** intermediates during the first stage, which were obtained and characterized earlier by the authors [116,117]. Intermediate amino-succinimides **130** through the formation of Schiff bases **131** are converted into a bisamide product without the use of additional synthetic procedures under reductive amination conditions. Moreover, the process already proceeds at room temperature. However, lactam products **132** and **133** were obtained as a mixture of cis/trans isomers in the relative configuration of substituents on the lactam ring.

## 6. Conclusions

As can be seen from the presented review, heterocyclic systems based on *N*-arylmaleimides have attracted close attention from researchers for a long time. The interest in the chemical transformations of maleimides is determined by the presence of several reaction centres and the possibility for the synthesis of heterocyclic systems with a wide range of biological effects, including drugs based on them. In this review, we have tried to systematize all currently available data on the products and features of the interaction of *N*-arylmaleimides with various binucleophiles. However, it should be noted that most of the efficient synthetic pathways lead to partially saturated mono- and bicyclic heteroatomic (PSBH) frameworks with an increased content of C*sp*^3^ atoms. The presented data can help in understanding and expanding the chemistry of *N*-arylmaleimides, in particular, in identifying new directions for their application in the synthesis of various non-aromatic heterocyclic systems.

## Data Availability

Not applicable.

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
