# Peer review of "Recyclization of Maleimides by Binucleophiles as a General Approach for Building Hydrogenated Heterocyclic Systems"

_molecules, 2022, doi:10.3390/molecules27165268_

Round 1
Reviewer 1 Report
Type of manuscript: Review
Title: Recyclization of maleimides by binucleophiles as a general approach for building hydrogenated heterocyclic systems
Special Issue: Recent Advances in Cyclization Reactions
Comments: In this review the authors have described synthesis of various saturated heterocyclic systems, frequently encountered fragments in many pharmacologically active compounds, starting from inexpensive N-substituted maleimides. Maleimides being one of the privileged heterocyclic building blocks with multiple reactive sites provides great avenues to generate numerous useful bioactive molecules, bioconjugate systems and polymers. However, the authors here focussed their discussion on existing synthesis of saturated heterocyclic systems based on recyclization of N-substituted maleimides by different heteroatoms containing di and ploy-nucleophiles.
Here, the author tried to give a comprehensive view about a comparatively less covered topic in maleimide chemistry covering the literature of several decades. However, though these perspectives have not been presented earlier in such focussed manner, the chemistry is well known and has been frequently discussed in many other related topics on heterocyclic chemistry. Thus, the review does not discuss the recent advances on cyclization reaction. The manuscript looks more like a systematic summary of related reported works. The correlation and differences among the reactivity of different sites of maleimides towards different nucleophiles and consequent outcomes are not adequately discussed. Though the structural diversity aspects are systematically presented, this manuscript has missed in providing mechanistic insight/scientific import and highlighting impactful future perspective or direction of this work.
Therefore, in its current form I cannot recommend the publication of this manuscript in Molecules. However, considering the distinctiveness of the review topic, a substantially rewritten manuscript, highlighting scientific insights and future potential/direction, can be considered for publication in Molecules or other allied sister journals.
Author Response
Dear Reviewer,
Thank you for the thorough analysis of our article. We understand your point of view regarding the general idea of the article. However, we are afraid we do not quite agree with it. You pointed out that the article does not discuss the latest advances in the cyclization reactions of maleimides. In our review, we did not aim to detail the development prospects of the chemistry of this preferential heterocyclic building block. The purpose of our study was to classify the existing inhomogeneous data on the processes of recyclization of N-substituted maleimides with various linear and cyclic di- and polynucleophilic agents as a general method of building promising biologically active hydrogenated polycyclic systems. The article also focuses on the dependence of the structure of the formed compounds on the structure of the nucleophilic component and the chosen conditions. We believe that our article provides the readers with a brief overview of the main advances in the aforementioned fields and demonstrates that they can be effectively used in strategies for heterocyclic synthesis.
Reviewer 2 Report
molecules-1776757
Review1
Recyclization of maleimides by binucleophiles as a general approach for building hydrogenated heterocyclic systems
Dmitriy Yu. Vandyshev1,*, Khidmet S. Shikhaliev1 4
This review is focused on the synthesis of hydrogenated heterocycles by means of the use of maleimides and their reactions with binucleophiles. The discussion of the literature data is well organized and divided according to the different nature of the nucleophiles , i.e. 1,3-N,N- and 1,4-N,N-dinucleophiles, 1,3-N,S-dinucleophiles and 1,4-N,S-binucleophiles, 1,4-N,O-binucleophiles, 1,3- C,N-binucleophiles and polynucleophilic reagents.
Hence I support the acceptance of this work with minor revisions:
1) lines 68-72
“An analysis of the available literature data to draw a general conclusion about the sequence of these reactions: the initial nucleophilic addition of a di- or polynucleophile according to the Michael reaction to the activated multiple bond of the imide and subsequent recyclization of the intermediate succinimide intermediate due to intramolecular nucleophilic substitution with the participation of another nucleophilic centre and one of the carbonyl groups”.
The sentence is not clear
2) Table 1. : are the compounds numbers 10/11 correct? Or should be 15/16??
Author Response
Dear Reviewer,
Thank you for the thorough analysis of our article. We have corrected all typos in the text and rephrased the sentence that you found incomprehensible.
Reviewer 3 Report
Recyclization of maleimides by binucleophiles as a general approach for building hydrogenated heterocyclic systems.This review summarizes the progress in the interaction of N-arylmaleimides with various binucleophiles.The interest in the chemical transformations of maleimides is determined by the presence of several reaction centres and the possibility of synthesis of heterocyclic systems with a wide range of biological effects, including drugs, based on them. The presented data can help in understanding and expanding the chemistry of N-arylmaleimides, in particular, in identifying new directions of their application in the synthesis of various non-aromatic heterocyclic systems.
The review is of good scientific quality and the rich and instructive graphic realizes the understanding of the obtained results and of their significance. The manuscript needs no language and grammar corrections. The manuscript is written straight forward.
With the great development of acceptor-, acceptor-acceptor-, donor-, donor-donor- and donor-acceptor- carbene chemistry, new carbene precursors and new methods involved in organic synthesis have been emerging.
The study is a meaningful suppliment to the series of publications regarding the heterocyclic compounds (with P, S, N atoms) and their complexes with transition metals related to natural products: synthesis, structural analysis and investigation of their biological activity, that have been extensively studied because their important properties and applications, especially in biological activities, such as, anti-microbial, anti-proliferative (prostate cancer cells), anti-cancer , anti-influenza and with antioxidant activity. In Introduction the autors did not reflect any other field of another heterocyclic with the important applications as chiral ligands for metal catalyst or receptors especially in biological activities.
Because the authors have been presented in the References part of the manuscript a series of scientific papers, that describe a lot of other derivatives, respectively their biological activities, the authors well have also to present in the introduction part the data about the other heterocyclic aromatic amines, heterocyclic phosphorous compounds or organometallic complexes.
Still I believe that you should describe in the introduction more generally synthesis of heterocyclic compounds. It is of interest for synthetic chemists which wide use have heterocyclic compounds (with P, S, N atoms). Examples of relevant publications are given below. It is recommended to the authors to cite these papers to give their introduction a wider base.
Synthesis, structure, and reactivity of tetrakis(o,o-phosphorus)-bridged calix[4]resorcinols and their derivatives, Vollbrecht, A., Neda, I., Thonnessen, H., Jones, P.G., Harris, R.K., Crowe, L. A. Schmutzler, R., Chemische Berichte, 1997, 130, 1715-1720
N-Heterocyclic Carbenes (NHC) Derived from Imidazo[1,5-a]pyridines Related to Natural Products: Synthesis, Structure and Potential Biological Activity of Some Corresponding Gold(I) and Silver(I) Complexes, Monica Mihorianu, M. Heiko Franz2,4, Peter G. Jones, Matthias Freytag1 Gerhard Kelter, Heinz-Herbert Fiebig, Matthias Tamm*and Ion Neda * Appl. Organometal. Chem. 2016, 30, 581-589
Some references should be inserted.
In conclusion of my review
I recommend this manuscript for publication with minor revisions!
Author Response
Dear Reviewer,
Thank you for the thorough analysis of our article. We would like to point out that in the introduction we consider the importance of building completely or partially hydrogenated heterocyclic systems. We point out that it is the recyclization of N-substituted maleimides with various binucleophiles that can be considered a general method of building promising biologically active hydrogenated polycyclic systems. The articles you suggested, as well as similar ones, focus on the synthesis of aromatic compounds, which is not in line with the idea of our article.
Round 2
Reviewer 1 Report
The authors have not made any significant modification in the manuscript to make it more impactful by incorporating vital mechanistic insights and discussing its relevance to synthetic or medicinal chemistry. Therefore, I do not recommend publication of this manuscript in Molecules.
Author Response
Dear reviewer no. 1, thank you for the re-review.
After careful analysis of your comments, two main points have been identified:
1) Lack of discussion on the topic regarding advances in cyclization reactions.
2) Lack of correlation and differences in the reactivity of arylmaleimides.
We would like to comment on them as follows:
1) Due to the fact that cyclization reactions are a very large area of organic chemistry, we would like to get specifics about what achievements in this area we should write about? We believe that within the framework of the narrow topic of this review, we fully reflect all currently available information in the literature.
2) We would like to note that "aspects" and "correlation" are in a certain sense equivalent and, ultimately, this remark with your subsequent discussions contradict themselves. In addition, in our review, we only analysed the literature data and reflected it in all the published dependences of the structure of the obtained compounds on the nature of binucleophiles. Data on the difference in reactivity determined by the structure of the arylmaleimides were not shown in the analysed data.